# Assessment of hospitalization costs and its determinants in infants with clinical severe infection at a public tertiary hospital in Nepal

**Suchita Shrestha[1][¤], Ram Hari Chapagain[2], Debjani Ram Purakayastha[3], Srijana Basnet[4], Nitya Wadhwa[3], Tor A. Strand[5,6], Sudha Basnet[4,5]***

**1** Department of Pediatrics, Institute of Medicine, Child Health Research Project, Tribhuvan University, Kathmandu, Nepal, **2** Medical Department, Kanti Children's Hospital, Kathmandu, Nepal, **3** Pediatric Biology Centre, Translational Health Science and Technology Institute, Faridabad, Haryana, India, **4** Department of Pediatrics, Institute of Medicine, Tribhuvan University, Kathmandu, Nepal, **5** Centre for Intervention Science in Maternal and Child Health, Centre for International Health, University of Bergen, Bergen, Norway, **6** Department of Research, Innlandet Hospital Trust, Lillehammer, Norway

¤ Current address: Oxford University Clinical Research Unit, Patan Academy of Health Sciences, Kathmandu, Nepal

* sudhacbasnet@gmail.com

**Data Availability Statement:** Data available on request. In order to meet ethical requirements for the use of confidential patient data, requests must

## Abstract

Sepsis, an important and preventable cause of death in the newborn, is associated with high out of pocket hospitalization costs for the parents/guardians. The government of Nepal's Free Newborn Care (FNC) service that covers hospitalization costs has set a maximum limit of Nepalese rupees (NPR) 8000 i.e. USD 73.5, the basis of which is unclear. We aimed to estimate the costs of treatment in neonates and young infants fulfilling clinical criteria for sepsis, defined as clinical severe infection (CSI) to identify determinants of increased cost. This study assessed costs for treatment of 206 infants 3–59 days old, enrolled in a clinical trial, and admitted to the Kanti Children's Hospital in Nepal through June 2017 to December 2018. Total costs were derived as the sum of direct costs for bed charges, investigations, and medicines and indirect costs calculated by using work time loss of parents. We estimated treatment costs for CSI, the proportion exceeding NPR 8000 and performed multivariable linear regression to identify determinants of high cost. Of the 206 infants, 138 (67%) were neonates (3–28 days). The median (IQR) direct costs for treatment of CSI in neonates and young infants (29–59 days) were USD 111.7 (69.8–155.5) and 65.17 (43.4–98.5) respectively. The direct costs exceeded NPR 8000 (USD 73.5) in 69% of neonates with CSI. Age <29 days, moderate malnutrition, presence of any sign of critical illness and documented treatment failure were found to be important determinants of high costs for treatment of CSI. According to this study, the average treatment cost for a newborn with CSI in a public tertiary level hospital is substantial. The maximum limit offered for free newborn care in public hospitals needs to be revised for better acceptance and successful implementation of the FNC service to avert catastrophic health expenditures in developing countries like Nepal. Trial Registration: CTRI/2017/02/007966 (Registered on: 27/02/2017).

be approved by the Nepal Health Research Council (NHRC) and the Regional Committee for Medical and Health Research Ethics in Norway. Requests for data should be sent to the authors, by contacting NHRC (http://nhrc.gov.np), or by contacting the Department of Global Health and Primary Care at the University of Bergen (post@igs.uib.no).

**Funding:** The work described in the manuscript was supported by a writing grant awarded by the Centre for Intervention Science in Maternal and Child Health (CISMAC; project number 223269), which is funded by the Research Council of Norway through its Centres of Excellence scheme and the University of Bergen (UiB), Norway. Authors SS and RHC were direct recipients of the writing grant. The funders had no role in study design, data collection and analysis, decision to publish, or preparation of the manuscript.

**Competing interests:** The authors have declared that no competing interests exist.

## Introduction

In 2018, of the estimated 5.3 million deaths in children under 5 years of age, 2.5 million (47%) occurred in the first month of life [1]. Neonatal disorders are one of the leading causes of sepsis related deaths [2]. Sepsis in the newborn is an important and preventable cause of death [3]. Moreover, newborns treated for sepsis are at an increased risk of adverse neurodevelopmental outcomes, which adds to the economic burden to families in developing countries [4–7]. The newborn mortality rate of 21 per 1000 live births, contributes to two thirds of all infant deaths in Nepal [8]. While sepsis is a major cause of neonatal death [9], reported burden of neonatal sepsis varies widely, ranging from 2% to 32%, in the few studies that have been conducted in health facilities across Nepal [10,11].

Amongst the microorganisms causing neonatal sepsis, bacteria are the most common [12]. The lack of an internationally accepted consensus definition of neonatal sepsis, non-specific signs and symptoms, having to wait for 48–72 hours for results of blood culture, including the low sensitivity of this test poses challenges for the early detection and successful management of this condition [13]. Therefore, after collecting samples for blood culture and other tests, treatment with antibiotics is initiated in all clinically suspected cases of neonatal sepsis. The World Health Organization (WHO) recommends hospitalization and treatment of infants less than 2 months of age identified with signs of possible serious bacterial infection. [14,15]. The criteria we used to identify young infants with sepsis in this study is an adaptation of the signs by WHO that we defined as clinical severe infection (CSI), to increase the specificity of the diagnosis [16].

In 2016, Nepal spent 6.3% of GDP on health [17] and out of pocket spending as a percentage of total health expenditure was 55.4% [18]. While voluntary community-based health insurance plans have been introduced in 36 districts of Nepal since 2018 [19], their coverage remains sporadic and there is no other publicly run health insurance plan in the country. The Government of Nepal (GoN) introduced the Free Newborn Care (FNC) service in 2015–2016 [19,20]. The FNC service, targeting sick neonates requiring inpatient care in public health facilities, aims to make health care provision equitable by preventing inaccessibility due to poverty. It has been designed to disburse costs, in three set packages A, B and C, according to type of health facility and service provision [20]. The amount specified in each package is expected to cover costs for investigations, medicines and bed charges for the duration of stay in hospital. Package A (NPR 1000) covers costs for newborn care at birthing centres, Package B (NPR 2000) for care provision in Special Newborn Care Units and Package C (NPR 5000) for care in the Neonatal Intensive Care (NICU). Tertiary level public hospitals can claim a maximum of NPR 8000 (Package A + B + C) for services it is likely to provide to a hospitalized sick neonate and included in all three packages [20]. While this is an important initiative to increase coverage to include marginalized groups, the basis for fixing the cost at NPR 8,000 is unclear. In a recent study on the status of FNC services, the most common cause of hospital admission was neonatal sepsis, and implementation of the program was delayed in many health facilities as the maximum limit of NPR 8000 offered was insufficient for providing tertiary level care to a sick newborn [21].

Among the studies done worldwide on costs related to sepsis in infants only a few are from LMIC settings [22,23] and Nepal [24,25]. In a clinical trial, assessing the efficacy of zinc for the treatment of clinical severe infection (CSI) in infants aged 3–59 days, we also documented the out-of-pocket treatment costs of parents/guardians at the study site in Nepal [16]. The knowledge on the costs may be important for the successful implementation of the FNC service by the GoN. This study aims to generate the information by estimating costs of treatment of CSI in infants and to identify determinants of increased costs.

## Methods

The study site, Kanti Children's Hospital (KCH) in Kathmandu, was part of a double-blind randomized placebo-controlled multicentre trial. KCH, is a tertiary, government referral hospital for children in Nepal.

Study participants were infants, 3–59 days of age, fulfilling criteria for CSI and requiring hospital admission. We adapted the inclusion criteria from WHO IMCI [14] and IMNCI [15] to identify very sick infants with CSI [26]. Eligibility criteria used to enroll study participants are outlined in Fig 1. The enrolled infants were admitted in the hospital, administered the study medication (zinc or placebo dispersible tablets) in two divided doses of 5 mg each for a period of 14 days, treated using standard case management protocols and followed up till discharge or other study outcomes, death and leaving hospital against medical advice. Approval was obtained from Nepal Health Research Council (NHRC) and institutional ethics committees. Written informed consent was obtained from parents/guardians of eligible infants prior to enrolment. Details of trial procedures are outlined in the study protocol [16]. The study was conducted over a period of 18 months from 26[th] June 2017 to 31[st] December 2018.

As part of the study procedures, direct hospital costs (bed charge, investigation and medicine cost) for each enrolled infant were reimbursed to the participant's parents/guardians.

We derived total cost of treatment for CSI as a sum of direct and indirect cost as shown in Fig 2. To calculate direct cost, the amount in the medical bills that were collected from the parents/guardians of each infant was entered in a database under the headings shown in Fig 2. Indirect cost, defined as work time loss of parents [27], was computed based on several assumptions. We assumed work time loss for a single parent/guardian and used length of hospital stay as a proxy for time lost. We input minimum daily wages for calculating indirect costs. The minimum daily wage of NPR 517 was taken from the notice published in the Nepal Gazette for the year 2018 by the Ministry of Labor [28].

The bed charges at KCH vary by ward. We kept a record of where study infants were admitted. While most study infants were admitted to Neonatal Intermediate Care Unit (NIMCU), some had to be transferred to Neonatal Intensive Care Unit (NICU) and some admitted in

---

**Inclusion** (signs of clinical severe infection):

- Stopped feeding well (where poor feeding was assessed by observation)
- Severe chest in drawing
- Axillary temperature ≥38.0˚ C or <35.5 ˚C
- Movement only when stimulated

**Exclusion**:
- Surgical or other conditions that interfere with oral/nasogastric feeding on admission
- Requiring surgical intervention
- Documented evidence of having received zinc in the last 48 hours
- Documented evidence of having received injectable antibiotics for ≥48 hours before admission
- Weight for age≤-4.5z

An infant of age 3 – 59 days, having been well at some point from birth till current episode of illness with any **one** inclusion and no exclusion criterion was eligible for enrolment into the study

**Fig 1. Eligibility criteria for study participants.**

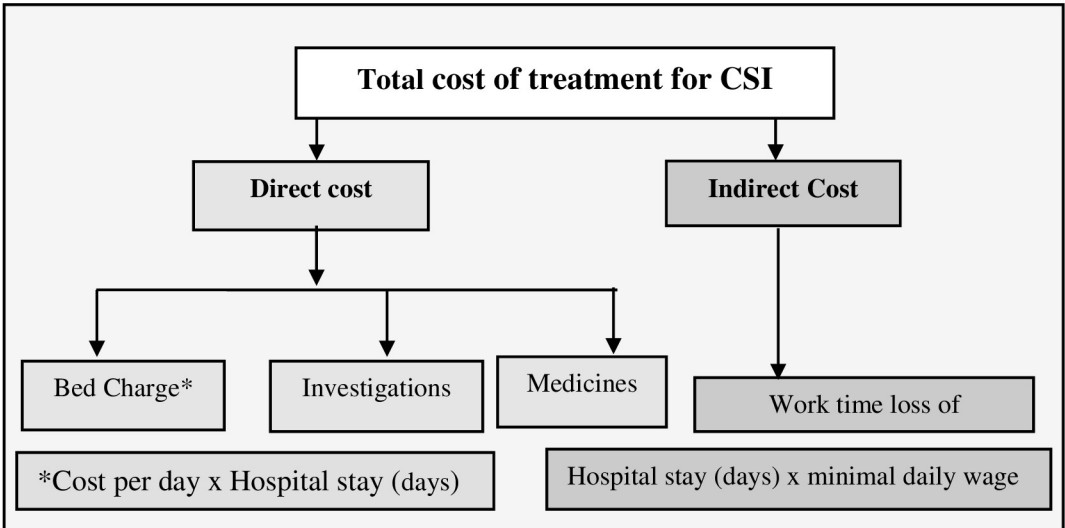

**Fig 2. Components of treatment costs for clinical severe infection in infants aged 3–59 days.**

medical ward, paying ward and private cabins. All cost data collected in Nepali Rupees (NPR) have been converted to US dollar using the average exchange rate for 2018 (NPR 108.9 = USD 1) for the purpose of comparison with other similar studies [29].

## Statistical analysis

The demographic and clinical data of study participants was entered in Microsoft Access database and analysed using STATA software (version 15). Means with standard deviations and 95% confidence intervals (CI); and medians with interquartile range (IQR) were used to describe quantitative variables and proportions for categorical variables. Infants were dichotomized in age groups: neonates (age<29 days) and young infants (age ≥29 days). The direct cost for treatment of CSI was dichotomized into categories of ≥ NPR 8000 and <NPR 8000.

Multivariable linear regression models were developed to identify determinants of cost. For both the dependent variables, total and direct cost, we included the following as candidate variables in the multivariable regression analyses: Age group (neonates vs. young infants), gender, nutritional status, place of birth, mother's age, mother's education, father's education, symptoms of illness (fever, lethargy, stopped feeding well), signs of clinical severe infection (severe chest indrawing, febrile or hypothermia, movement only when stimulated), signs of critical illness (grunting, nasal flaring, convulsions and no movement at all) and treatment failure. Collinearity was assessed by calculating the variance inflation factor [30]. We first assessed the crude associations of relevant independent with dependent variables i.e. total and direct costs using linear regression. Variables with p< 0.20 were included in the multivariable models and those variables which were still significant, i.e., being associated with a p-value of < 0.05, were retained in the model. In these models we included the other variables one at a time and kept them if significant, in a manual stepwise approach outlined by Hosmer and Lemeshow [30].

## Results

Through June 2017 to December 2018, 213 infants aged 3–59 days were enrolled in the clinical trial at KCH. Seven children, in whom at least one component of cost data was missing, were excluded from the analysis. Among the 206 infants, 138 (67%) were neonates and admitted to

neonatal units. Among the young infants (N = 68), 18 (26%) were admitted to Medical ward with no bed charge and others (74%) were admitted to Paying wards and private cabins. All the 206 infants were retained in the analysis. Demographic and clinical characteristics of neonates and young infants are outlined in Table 1. The demographic characteristics of both the groups were similar. More than half of the neonates (52%) presented with complaint of fever and among them almost two third were febrile (axillary temperature $\geq 38\,^{\circ}$C) at enrolment. Severe chest indrawing was the most common sign of CSI in young infants (87%). The median (IQR) length of hospital stay for neonates and young infants was 7 (5–11) and 7 (5–9) days respectively. Treatment failure, defined as either initiation of life support or change of antibiotics due to persistence/appearance/reappearance of signs of CSI or death [16] was documented in 22% of neonates and 12% of young infants. Coagulase negative staphylococcus was isolated in most blood cultures using BACTEC in neonates.

Cost of bed charge accounted for the largest component of direct costs in both neonates and young infants (Fig 3). The mean (95% CI) direct cost for treatment of CSI in neonates and young infants was NPR 15096(13099–17092) and NPR 9698 (7579–11816) respectively (S1 Table). The median total costs and its components are described in Table 2. The direct costs of treatment exceeded NPR 8000 in 69% neonates with CSI.

As the distribution of the outcome variables (direct and total cost) was skewed, we log transformed the data before we fitted the regression models. The coefficients of the dependent variables, in both univariable and multivariable regression models, were exponentiated to enable interpretation of the results. The association between the direct cost and the independent variables is depicted in Table 3.

Age <29 days (neonates), moderate malnutrition and presence of any sign of critical illness at the time of admission were independently associated with direct costs for treatment of CSI (Model 1). While treatment failure was also independently associated with direct costs, addition of this variable did not change the association between the outcome and other independent variables (Model 2) as depicted in Table 3. The results of the multivariable regression analysis with total cost as the outcome variable were very similar and is available in S2 Table.

## Discussion

In this study we report an analysis of costs for treatment of CSI from data collected in 206 participants enrolled in a clinical trial assessing the efficacy of zinc in hospitalized infants that were 3–59 days old. The average direct costs for the treatment of CSI in neonates in a public tertiary hospital in Nepal was NPR 15096 (USD 138.6) in 2018 which was higher in comparison to young infants. After accounting for inflation, the estimated average direct costs would be NPR 16741 in 2020 [31]. The bed charge for hospital stay was a major component of direct cost for treatment of CSI in both groups. In the multivariable regression model, age <29 days (neonates) and moderate malnutrition was associated with incremental costs of 61% and 33% respectively. Similarly, the presence of signs of severe illness and experiencing treatment failure was associated with increment of 26% and 100% direct costs respectively.

Two thirds of the study participants were neonates (3–28 days). The direct costs of treatment exceeded the maximum limit of NPR 8000 offered in the FNC service by the GoN in 69% in this group. Our finding of substantial treatment costs for care of a sick newborn supports the reason for reluctance of public tertiary level hospitals to initiate the FNC service, stating that the maximum amount allocated by GoN is insufficient [21].

The bed charges, the largest component of direct costs, depended on which ward the infant was admitted to and the length of hospital stay. Our study is similar to studies in India and Bangladesh [32,33], where the bed charges account for significant cost of treatment of

**Table 1. Baseline characteristics of study participants hospitalized with clinical severe infection at Kanti Children's Hospital.**

| Demographic characteristics | Neonates (N = 138) | | Young infants (N = 68) | |
|---|---|---|---|---|
| | N | Value | N | Value |
| Male (%) | 138 | 87 (63) | 68 | 43 (63) |
| Mean weight at presentation in grams (SD) | 138 | 3215 (722) | 68 | 3956 (920) |
| Moderately malnourished (Weight for age Z score ≤-2) (%)[a] | 138 | 38 (27) | 68 | 21 (31) |
| Severely malnourished (Weight for age Z score ≤-3) (%)[a] | 138 | 15 (11) | 68 | 15 (22) |
| Place of delivery | 138 | | 68 | |
| Health facility (%)[b] | | 116 (84) | | 56 (82) |
| Home (%) | | 22 (16) | | 12 (18) |
| Mean age of mother (SD) | 136 | 24 (4) | 66 | 24 (6) |
| Mean age of father (SD) | 136 | 28 (1) | 66 | 27 (5) |
| Education of parents [c] | 136 | | 67 | |
| Mothers with no formal education (%) | | 20 (15) | | 8 (12) |
| Mothers with primary education (%) | | 26 (19) | | 11 (16) |
| Mothers with education above primary (%) | | 90 (66) | | 48 (72) |
| Fathers with no formal education (%) | | 10 (7) | | 5 (7) |
| Fathers with primary education (%) | | 31 (23) | | 20 (30) |
| Fathers with education above primary (%) | | 95 (70) | | 42 (63) |
| Unemployed mothers (%)[d] | 138 | 124 (90) | 68 | 56 (82) |
| Unemployed fathers (%) | 137 | 14 (10) | 67 | 3 (4) |
| **Clinical characteristics** | | | | |
| Symptoms | | | | |
| Fever (%) | 138 | 72 (52) | 68 | 22 (32) |
| Stopped feeding well (%) | 138 | 44 (32) | 68 | 10 (15) |
| Lethargy (%) | 138 | 39 (28) | 68 | 7 (10) |
| Diarrhoea (%) | 138 | 15 (11) | 68 | 5 (7) |
| Signs | | | | |
| Severe chest indrawing (%) | 138 | 74 (54) | 68 | 59 (87) |
| Febrile (%)[e] | 137 | 47 (34) | 68 | 12 (18) |
| Movement only when stimulated (%) | 138 | 35 (25) | 68 | 6 (9) |
| Nasal flaring (%) | 138 | 16 (12) | 68 | 15 (22) |
| Grunting (%) | 138 | 4 (3) | 68 | 4 (6) |
| Convulsions (%) | - | - | 68 | 1 (1) |
| No movement at all (%) | - | - | 68 | 1 (1) |
| Any sign of critical illness (%) [f] | 138 | 19 (14) | 68 | 19 (28) |
| Median length of hospital stay (IQR) | 138 | 7 (5–11) | 68 | 7 (5–9) |
| Blood culture positive (%) | 138 | 29 (21) | 68 | 14 (21) |
| **Treatment failure (%) [g]** | 138 | 31 (22) | 68 | 8 (12) |
| Requiring initiation of life support [h] (%) | 31 | 2 (6) | 8 | 1 (12) |
| Requiring change in antibiotics (%) | 31 | 29 (93) | 8 | 7 (87) |

(*Continued*)

**Table 1.** (Continued)

| Demographic characteristics | Neonates (N = 138) | | Young infants (N = 68) | |
|---|---|---|---|---|
| | N | Value | N | Value |
| Death during hospitalization (%) | 31 | 1 (3) | 8 | 1 (12) |

[a] Calculated using WHO growth standards.

[b] Includes delivery at hospital and other health facilities.

[c] Primary education is up to grade 6 and higher is > 6 grade.

[d] Unemployed or housewives.

[e] Axillary temperature $\geq$ 38°C.

[f] Having nasal flaring OR grunting OR convulsions OR no movement at all.

[g] Initiation of life support or change of antibiotics due to persistence/appearance/reappearance of signs of CSI or death.

[h] On mechanical ventilation or vasoactive drugs.

hospitalized children. Neonates at KCH are admitted to the NIMCU and shifted to NICU if they require advanced support. The bed charges in both units are higher than that of other wards. Amongst the 22% of neonates with treatment failure, 93% required a change in antibiotics, and 6% were transferred to NICU; factors likely to be associated with a longer duration of hospital stay. Moreover, the study protocol required that participants be kept in hospital for an additional period of 48 hours following disappearance of clinical signs of illness to fulfil

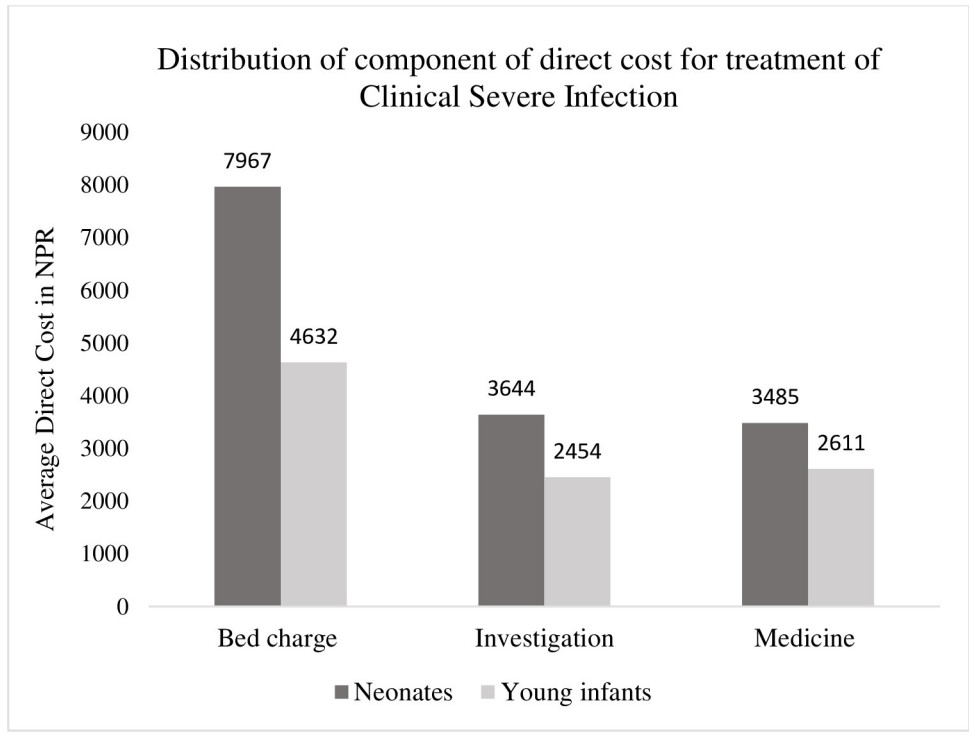

**Fig 3. Distribution of components of direct costs for treatment of clinical severe infection in study participants admitted to Kanti Children's Hospital.**

**Table 2. Median costs for treatment of clinical severe infection in study participants at Kanti Children's Hospital.**

| Cost of treatment for CSI | Median (IQR[a]) | | | |
| --- | --- | --- | --- | --- |
| | Neonates (N = 138) | | Young infants (N = 68) | |
| | NPR | USD* | NPR | USD* |
| Total cost | 17075 (9817–22233) | 156.8 (90.1–204.2) | 11569 (7292–15006) | 106.2 (67.0–137.8) |
| Indirect cost | 3619 (2585–5687) | 33.2 (23.7–52.2) | 3619 (2585–4653) | 33.2 (23.7–42.7) |
| Direct cost | 12162 (7602–16931) | 111.7 (69.8–155.5) | 7097 (4724–10730) | 65.2 (43.4–98.5) |
| Bed charge | 7400 (3950–9200) | 67.9 (36.3–84.5) | 3200 (200–6557) | 29.4 (0.01–60.2) |
| Investigation | 2567 (1820–3880) | 23.6 (16.7–35.6) | 1920 (1450–2710) | 17.6 (13.3–24.9) |
| Medicine | 2202 (1392–4472) | 20.2 (12.8–41.1) | 1792(1085–2888) | 16.5 (10.0–26.5) |

*1 USD = 108.9 NPR (Average in 2018) [29].

[a] IQR = Interquartile range.

criteria for defining recovery. This in addition to treatment failure resulting in prolonged stay might have contributed to the increased costs.

In a study done in Nepal by Sunny et al [25] exploring the Out Of Pocket Expenditure for treatment of sick newborns (0–28 days), costs ranged from USD 13 to 226 with a mean cost of USD 31.3. The median costs for newborns admitted with culture proven sepsis in the study was USD 25.8 (13.6–139.8) [25]. This finding is very different from our finding of median direct costs of USD 111.7 (69.8–155.5) in neonates for treatment of CSI. There could be several reasons for this disparity between the findings of the two studies. Our study was based in a tertiary level, referral hospital that provides inpatient service to only out born babies, whereas the study by Sunny et al was done in secondary level hospitals with facilities for care of inborn neonates. A study in India on costs of treatment for severe pneumonia in 2 to 36 month infants [34] also reports that the cost of hospitalization was higher in tertiary compared to secondary level hospitals. The median length of hospital stay of 3 days in the study done by Sunny et al was shorter in comparison to 7 days of the present study [25]. It is likely that infants with more severe illness were referred from the secondary level hospitals for higher level of care explaining the shorter duration of hospital stay and lower cost in study by Sunny et al when compared to the present study.

**Table 3. Determinants of direct cost of treatment for clinical severe infection in Nepali infants aged 3–59 days (N = 206).**

| Covariates | Crude Estimates | | | Adjusted Estimates | | | | | |
| --- | --- | --- | --- | --- | --- | --- | --- | --- | --- |
| | Exp[#] | 95%CI | p | Model 1 | | | Model 2 | | |
| | | | | Exp[#] | 95%CI | p | Exp[#] | 95%CI | p |
| Neonate | 1.68 | 1.4,2.0 | <0.001 | 1.76 | 1.5,2.1 | <0.001 | 1.61 | 1.4,1.9 | <0.001 |
| Moderately malnourished | 1.32 | 1.1,1.6 | 0.008 | 1.35 | 1.1,1,6 | 0.002 | 1.33 | 1.1,1.6 | 0.001 |
| Any sign of critical illness | 1.18 | 0.9,1.5 | 0.180 | 1.34 | 1.1,1.7 | 0.010 | 1.26 | 1.0, 1.5 | 0.023 |
| Treatment failure | 2.35 | 1.9,2.9 | <0.001 | | | | 2.14 | 1.8, 2.6 | <0.001 |

[#] Exponentiated coefficient of log transformed direct cost with 95% CI.

Model 1 (Demographic and clinical variables): Adjusted for gender, place of birth, mother's age, mother's education, father's education, fever, lethargy, stopped feeing well, severe chest indrawing and being febrile. Only significant associations shown. $R^2 = 0.18$.

Model 2 (Demographic and clinical variables and treatment failure): Adjusted for gender, place of birth, mother's age, mother's education, father's education, fever, lethargy, stopped feeing well, severe chest indrawing, being febrile and treatment failure Only significant associations shown. $R^2 = 0.35$.

The results from our study show that approximately one-third of the total cost is contributed by indirect cost which is similar to findings from studies done in India and Bangladesh [32,33]. The indirect cost does not include travel and living expenses and therefore the out-of-pocket expenditures of parents/guardians in the present study is likely to be higher. High out of pocket expenditure is likely to drive poorer households towards catastrophic health spending with adverse consequences on the household economy in low-income countries like Nepal [35].

Age <29 days (neonates), nutritional status of infants, presence of sign of severe illness and treatment failure were independent determinants of direct and total costs. All these determinants are closely linked to disease severity and duration of hospital stay that may explain their association with increased costs. Scrimshaw et al. were among the first to describe the vicious cycle between malnutrition and infection [36]. A more recent review reconfirms this association whereby malnourished children in addition to having increased frequency of infectious disease are also at significantly higher risk of more severe disease [37]. The association between younger age of infants with severe outcomes and prolonged time to recovery has also been identified in other studies [38,39]. A study done in Vietnam also demonstrated an association of younger age with increased cost for treatment of illnesses [22].

In this study, we estimated expenditures for treatment of CSI in infants using timely collection of receipts with entry of data at the earliest, which indicates that our results of direct costs are highly reliable. This study is nested in a randomized clinical trial that necessitated strict adherence to protocol for the inclusion, enrolment, management, and discharge of study participants with close monitoring of enrolled infants by the study physicians and senior paediatricians that were part of the team.

This study also has few limitations. We were able to derive accurate estimates of direct medical costs and not indirect costs. We did not collect information on family income, salaries of working parents, time parents spent away from work, expenses for travel and living costs of caregivers which limited our ability to estimate out of pocket expenditure and catastrophic health expenditure (CHE), both of which have long term economic implications for poorer households that are more likely to seek care in public hospitals in Nepal. The study protocol we followed restricted the participation of infants with severe illness that could not be fed orally or required transfer for advanced care such as mechanical ventilation and parenteral vasoactive drug support. The infants we excluded are most likely those with even higher costs for treatment that we could not account for. The study findings are limited to settings providing tertiary care in public health facilities, in Nepal. We provide data on costs that are limited to reimbursed expenses for treatment of CSI during hospital stay. We have not included information on costs for human resource wages, capital items and administration/support services etc. which would have been a better assessment of the economic burden to parent of an infant with CSI [40]. Despite this limitation the average direct cost for treatment of CSI is still higher than the maximum amount offered in the FNC service.

In this secondary analysis of data collected during a clinical trial, treatment costs for CSI in a public tertiary hospital exceeded the maximum limit of NPR 8000 in two thirds of neonates. This finding highlights the need of health policy makers in Nepal to consider increasing the limit for allocation of costs for provision of free newborn services in public hospitals. Meanwhile in order to contribute more towards evidence for out-of-pocket expenditure and CHE, prospective studies are recommended at different levels of health care delivery to accurately estimate both direct and indirect costs and inform policy.

## Supporting information

**S1 Table. Arithmetic mean cost for treatment of clinical severe infection in study participants at Kanti Children's Hospital.**
(TIF)

**S2 Table. Determinants of total cost of treatment for clinical severe infection in young Nepali infants aged 3–59 days (N = 206).**
(TIF)

**S3 Table. Bed charges for admission in different units at Kanti Children's Hospital.**
(TIF)

## Acknowledgments

We are grateful to all young infants and their families who took part in the clinical trial. We are indebted to our study assistant, Madhu Sudan Kuinkel for maintaining a record of caregiver costs, the study physicians, and other members of the Zinc Sepsis Study Group for their contribution to the study, staff of the Child Health Research Project and Kanti Children Hospital for their invaluable support. We are also grateful to Dr Jose Martinez for his valuable input and advice to improve the conduct of the study and the manuscript and to Centre for Intervention Science in Maternal and Child Health (CISMAC) for their support.

## Author Contributions

**Conceptualization:** Suchita Shrestha, Ram Hari Chapagain, Sudha Basnet.

**Data curation:** Suchita Shrestha, Ram Hari Chapagain, Tor A. Strand.

**Formal analysis:** Suchita Shrestha, Ram Hari Chapagain, Debjani Ram Purakayastha, Srijana Basnet, Nitya Wadhwa, Tor A. Strand, Sudha Basnet.

**Funding acquisition:** Nitya Wadhwa, Tor A. Strand, Sudha Basnet.

**Methodology:** Suchita Shrestha, Ram Hari Chapagain, Debjani Ram Purakayastha, Srijana Basnet.

**Project administration:** Nitya Wadhwa, Tor A. Strand, Sudha Basnet.

**Supervision:** Ram Hari Chapagain, Srijana Basnet, Tor A. Strand, Sudha Basnet.

**Writing – original draft:** Suchita Shrestha, Debjani Ram Purakayastha, Sudha Basnet.

**Writing – review & editing:** Suchita Shrestha, Ram Hari Chapagain, Srijana Basnet, Nitya Wadhwa, Tor A. Strand, Sudha Basnet.

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
