## [Decision Letter · Decision Letter 0]

9 Jun 2021

PONE-D-20-32967

Assessment of hospitalization costs and its determinants in young infants with sepsis at a tertiary hospital in Nepal

PLOS ONE

Dear Dr. Basnet,

Thank you for submitting your manuscript to PLOS ONE. After careful consideration, we feel that it has merit but does not fully meet PLOS ONE’s publication criteria as it currently stands. Therefore, we invite you to submit a revised version of the manuscript that addresses the points raised during the review process.

The reviewers have identified a number of aspects of the methodology and analysis that need considerable clarification. Please ensure that you respond thoroughly to all of the reviewers' comments when preparing your revised manuscript.

We look forward to receiving your revised manuscript.

Kind regards,

Jamie Males

Staff Editor

PLOS ONE

Journal Requirements:

**2.** Please ensure you have included the registration number for the clinical trial referenced in the manuscript."

3. Thank you for stating within the ethics statement "This manuscript is a secondary analysis of data collected during implementation of a clinical trial for which ethical approval was provided by the Nepal Health Research Council". Within the manuscript text please provide additional information regarding informed consent, please ensure you have also stated whether you obtained consent from parents or guardians of the minors included in the study.

5. We note that you have stated that you will provide repository information for your data at acceptance. Should your manuscript be accepted for publication, we will hold it until you provide the relevant accession numbers or DOIs necessary to access your data. If you wish to make changes to your Data Availability statement, please describe these changes in your cover letter and we will update your Data Availability statement to reflect the information you provide

Reviewers' comments:

Reviewer's Responses to Questions

**Comments to the Author**

1. Is the manuscript technically sound, and do the data support the conclusions?

Reviewer #1: Yes

Reviewer #2: Partly

Reviewer #3: Partly

2. Has the statistical analysis been performed appropriately and rigorously? 

Reviewer #1: Yes

Reviewer #2: Yes

Reviewer #3: Yes

3. Have the authors made all data underlying the findings in their manuscript fully available?

Reviewer #1: Yes

Reviewer #2: Yes

Reviewer #3: No

4. Is the manuscript presented in an intelligible fashion and written in standard English?

Reviewer #1: Yes

Reviewer #2: Yes

Reviewer #3: Yes

5. Review Comments to the Author

Reviewer #1: In this study, the authors calculated both the direct and indirect costs of hospitalization for severe infection in young infants in Nepal and compared these costs to the maximum limit set by the Government of Nepal’s Free Newborn Care reimbursement. The study addresses an important topic and has been nicely done, and the paper is well-written. I do have some comments and questions for the authors.

Main comments:

1. My primary question concerns the commingling of sepsis with “clinical severe infection.” In the abstract the authors use the term “sepsis,” but in the Methods section state that the terms “sepsis” and “clinical severe infection” are interchangeable. While all sepsis infections are clinical severe infections, all clinical severe infections are not sepsis. Only 21% of the children in the study were culture positive for sepsis. There was, however, quite a difference in the median (IQR) costs for children with and without culture positive sepsis (ll.220-222).

It seems like it would be more accurate, then, to refer to these children as those with “severe infection, including sepsis” with some discussion about the infections of the 80% of children who were not culture-diagnosed with sepsis. In order to justify analyzing these two groups together, the authors should examine whether the two groups are actually similar enough (in characteristics and cost) to be lumped together or should be analyzed separately. Were children who met clinical criteria for sepsis negative by culture? If so, it bears discussing why the authors refer to all infections as “sepsis.”

To inform policy guidelines, the authors should at least do a sensitivity analysis of the two groups separately, since the Government decisions about reimbursements could at some point specify higher reimbursements for culture-diagnosed sepsis and not other non-sepsis infections (even though the costs are also high). I think that a more granular presentation of the data would be the most persuasive.

2. The other confusion is with the use of the word “infants” to refer sometimes to all the children in the study (l. 183; also, both age groups are labeled “infants” in Table 1), and sometimes only to the non-neonates (ll. 164-65, l.167). Sometimes the phrase “young infants” is used. In the Results section it is confusing what group constitutes the denominator of the reported percentages. The authors should use these terms consistently throughout the paper to avoid confusion, defining two groups (e.g., specifically defining “neonates” and “infants” as representing different age groups). At times these two groups are analyzed together and at times they are analyzed separately. Again, the authors should be consistent. If the two groups are found to be different enough not to be lumped together, they should be kept separate throughout. This might actually give more precise information than simply adjusting for age in the multivariable model. If the authors feel it is better to combine the groups, they should justify this decision.

Other comments:

Introduction:

The majority of the introduction (ll. 52-60) discusses infant deaths in general. This is interesting, but not really necessary. Discussing deaths in infants due to infection would be more relevant to the paper and make the introduction more focused.

l.68. Perhaps a different word could be used than “schemes” which has a somewhat negative connotation.

ll.108-113 Here the authors provide a nice and clear explanation of the calculation of indirect costs.

Results

• ll. 165-66 These costs are not normally distributed, so means and sds should not be reported. The correct summaries are in Table 2.

• l.167 Does “with sepsis” refer to all or only those with culture-positive sepsis diagnosis?

• ll.183-85 Please show the results of the analysis for the total costs. The authors’ inclusion of indirect costs (rightfully) in the analysis makes the total costs more relevant than the direct costs; not showing an analysis that includes them implies that they aren’t important (can show both if you want).

• I would argue, based on the R2 values, that the authors should use Model 2 as their final adjusted model.

Table 1.

• Refer to age groups by different names (“neonates” and “infants”) here and consistently throughout the paper.

• Education information would be more useful if parental information was combined. Two parents with above primary education are different from one parent with an above primary education and one with no education, or two with no education. I would suggest categorizing them into mutually exclusive groups: (1) neither parent has a formal education (2) one or both parents have a primary education (but no higher) and (2) one or both parents have an above primary education.

Table 3.

• What is the reference category for “Moderately malnourished?” (since this appears to be protective). You might want to reverse the reference group.

Discussion

• I am not sure where the USD 122 (l.198) comes from?

• In comparisons made with other studies, the authors should only compare parallel age groups. In ll.218-220, it sounds like all children (neonates and infants) are being compared to the Sunny study which enrolled only neonates. Again, I think it would be interesting to break out the two age groups in terms of characteristics (Table 1) unless the authors have tested whether there are no differences. The comparisons with other studies made in the discussion would also be less confusing with a more rigorous and consistent delineation between “neonates” and “infants.”

Overall, the authors have done a good job and address an important topic. I do feel that the paper would benefit from more granular analysis (neonates vs. infants; culture-dx sepsis vs. other infections) which might be more useful in informing Government policy. Also, I feel that it is misleading to define all infants as having sepsis, unless the hospital makes diagnoses of sepsis based on clinical diagnoses in the absence of culture-positive diagnosis.

Reviewer #2: Thank you for the opportunity to review the manuscript “Assessment of hospitalization costs and its determinants in young infants with sepsis at a tertiary hospital in Nepal” . My comments are:

In general, all abbreviations should be explained and the perspective of the costs should be mentioned explicitly according to the guidelines of economic evaluations.

Introduction

The introduction reflects the available knowledge about sepsis in the newborn and the importance of the study questions.

Methods

The term “caregivers” is a term of wide comprehension. For me is not clear what the authors mean: parents, relatives or even nurses?

It is not clear why the authors did not make a cost benefit or cost effectiveness analysis. I assume that the intervention was not successful. If I am right, this fact should be mentioned in the manuscript.

From my point of view the direct cost calculation is difficult to understand. What does is mean “investigation cost"? Lab cost, labor cost of physicians or what else? Furthermore, I am not sure that the indirect cost is calculated precisely. For example, can it be that newborns or infants were in the hospital for other reasons and sepsis occurred at any point if the hospital stay? This can lead to an overestimation of costs as well as the use of the hospital stay as a proxy for the time lost by the caregivers. Or are in the study only newborns and infants who are admitted to the hospital with diagnosis sepsis from different secondary level hospitals? I think this part of the manuscript needs some clarification for the reader. I am sorry but I am confused.

Statistical analysis

Statistical analysis is ok.

Results

For table 3 I am missing an explanation in the methods section. Furthermore, what is the rationale for using these models?

Discussion

In the discussion section it should be explained why the newest definition of sepsis was not used and why the medicine costs are lower than the investigation costs. That is surprisingly for me. Furthermore, the difference between charges and costs should be discussed. That ist not always the same.

Overall

This is interesting manuscript. Once again, that you for the opportunity to read it.

Reviewer #3: The present topic assesses the treatment costs for neonatal illnesses in tertiary care setting. Undoubtedly the topic is of great interest as healthcare decisions are largely dependent on costs and cost-effectiveness of services/technologies these days. At present there are several issues in the analysis and manuscript. First, the title of the study is ‘Assessment of hospitalization costs and its determinants in young infants with sepsis at a tertiary hospital in Nepal’ which I found a bit misguiding. In classical terms, cost and expenditures are two different concepts. In some instances, expenditures are proxy of costs but more often these are different. In this study authors have done an assessment of patient expenditures on hospital treatment for neonatal sicknesses to serve the purpose of informing FNC package rate which seems incorrect to me. Second, this study does not even comprehensively capture the information on patient expenditures to inform the economic burden of neonatal illnesses though the major heads have been covered. Third, the multivariable analysis is a bit unclear especially for the choice of variables theoretically selected for regression model and missing details how regression was performed.

I think there is a lot scope of improvement in the methodological and analytical part; and therefore, I recommend a major revision. Also, I suggest the authors should present this analysis as assessment of economic burden as a result of neonatal illnesses adding more data (if possible). Specific comments for each section are given below:

Introduction

• Page 3, lines 71-76: The description about the current reimbursement package under the insurance scheme is a bit unclear. Authors mention that the package is expected to cover costs such as investigations, medicines and bed charges for hospitalization and the maximum amount which can be claimed only in case of ICU which includes all A+B+C components. Is the package rate same for normal hospitalization and ICU? Is this package rate similar or differential across the public and private hospitals? Is there a scrutiny on claims raised by the hospitals? Any reflections from claims data about total claims settled under FNC package (or burn out ratio if possible)? More details would be useful to understand the context and to evaluate whether the current costing study serves the purpose of informing the required revisions in reimbursement rates of FNC package.

• Page 4, lines 78-79: “Moreover, this package does not take into account the indirect

out of pocket costs that caregivers of the sick neonates have to bear.” I doubt if we have any such precedence of package rates considering the indirect costs. Generally, the package rates tend to cover the treatment costs that too preferably the cost of provisioning of a particular service. In some instances, direct non-medical costs such as expenses for travel etc are covered but productivity losses are not covered as these neither fall within the scope of packages nor systems have enough resources to account for it.

Methods

• My main concern is authors study the patient expenses incurred for treatment of sick neonates in tertiary setting and claim that the aim of the present study is to inform the planning of budget and basis for revision in FNC package rate for neonatal care. I think the methodology employed is not appropriate to address the policy question in view of following arguments. Firstly, the patient expenditures for neonatal illnesses at one side are important to be assessed to understand the economic impact in terms of financial hardship to households but certainly not an appropriate basis for setting the prices of healthcare packages. Second, patient/household expenditures are subject to high variation and cost of medicines and diagnostics drives this variation. Even if the medicines, diagnostic services etc. were available from the hospital itself, the charges at public hospital are supposed to be subsidized and therefore, what patient pay does not adequately represent the cost of care in true economic terms. Moreover, the patients are not charged for time of human resource (Doctors, paramedical staff etc.) involved in treatment and care which accounts for around 40-70% of total cost of service and the capital resources consumed for service delivery. Third, the most appropriate way to inform such policy decisions is to estimate the value of resources consumed for delivery of a service that truly represent the cost of care. Now, this could be done employing the standard methodologies for healthcare costing (Bottom up, normative, mix methodology etc.). The normative costing approach seems to be perfect here as there are standard treatment protocols for neonatal care under IMNCI. There is some practical guidance available on the subject. (Özaltın A, Cashin C. Costing of health services for provider payment. A practical manual based on country costing challenges, trade-offs, and solutions. Arlington: Joint Learning Network for Universal Health Coverage. 2014; Translating Research to Policy: Setting Provider Payment Rates for Strategic Purchasing under India’s National Publicly Financed

Health Insurance Scheme)

• No information on perspective used for costing though it seems to be patients’ perspective which mismatches with policy question to be addressed.

• Statistically, median as a measure of average may be justified as it is least affected with high variations compared to mean but for practical purposes mean is a better choice specifically for reimbursement decisions. We must let the average get influenced by the variation (if this is genuine variation). The literature also suggest that the use of mean is quite popular in the costing studies specifically for informing budgetary and planning decisions.

• Human resource wages, cost of capital items and cost of administration/support services not accounted in estimation of costs no matter whether patients are paying for it or not if the overarching goal is to generate basis for FNC package rate. Its exclusion is only justified if authors claim the purpose of study to be assessment of patient out-of-pocket expenditures associated with neonatal treatment and care. But currently the latter argument does not hold completely true as the travel and other costs are absent.

• Authors have converted the costs from Nepalese rupee to USD using the exchange rate of 2018. As we are already in 2021, so I suggest authors to consider adjusting the estimates to current period for better utility.

• Page 6, lines 131-135: Authors have considered including following independent factors as determinants in multivariable regression Age of infant, gender, nutritional status, place of delivery, mother’s age, father’s age, mother’s education, father’s education, symptoms of illness (fever, lethargy, stopped feeding well), signs of clinical severe infection (severe chest indrawing, febrile or hypothermia, movement only when stimulated), signs of critical illness (grunting, nasal flaring, convulsions and no movement at all) and treatment failure.

In my opinion, the independent factors (as determinants/predictors) are included in the regression based on some theoretical relevance. Now, finding the association between these factors with outcome of interest does not guarantee the presence of any causal relationship which is a prerequisite in case of determinants. I doubt if there is any theoretical relevance of including factors like age of father and mother; place of delivery (what this stands for? Seems to be place of childbirth? If yes, what are categories within it? Home deliveries within this might be of greater importance). Also, there are too many syndromic variables. Why not to combine these and could be used as dummy variables. Otherwise, there will be a singularity issue in the regression even if the authors have taken care of multicollinearity issues.

• Page 6, lines 139-140: Please explain manual stepwise approach. I guess author meant guided stepwise regression. If yes, do we include or exclude the variables in regression at every step? Is it done for a single variable every time?

Results

• Page 8, Table 2: How come the average of independent components of costs i.e., bed charges, investigations and medicines not equal to average of total direct costs?

• I am not sure how the independent variables were introduced into the regression model. Some of variables seems to be on continuous scale whereas some of the variables are of nominal or ordinal nature such as place of delivery, education of parents, signs and symptoms etc. Use of dummy variables is generally recommended for multivariable regression in case of categorical variables. Please clarify.

• I think authors should have given the breakdown within each section i.e., medicines and investigations as it is important to understand what are the major drivers of costs within each subhead. For example, if a particular drug or a diagnostic test is predominantly determining the cost in respective subheads then there is a clear direction for policy makers for targeted action needed. This particular evidence will highlight the urge for strategic purchasing focusing on value-based pricing.

Discussion

The discussion part starts very well but losses the focus in between with too much comparison with other studies. I suggest the authors should curtail this part with most relevant comparisons. Also, the discussion should be revised in view of methodological comments given in the previous sections and should add a focused para on policy implications which should cover all the aspects suggested in previous comments.

Minor comments

• In the references 3, 6, 9 etc. the access date would be more relevant instead of date of citation.

• References 11, 12, 19 and 22 are Government reports, so the weblink and access date should also be provided.

• The WHO reference after reference 19 had not been numbered. Please check the in text citations also for consistency.

6. PLOS authors have the option to publish the peer review history of their article (what does this mean?). If published, this will include your full peer review and any attached files.

Reviewer #1: No

Reviewer #2: No

Reviewer #3: No

---

## [Author Response · Author response to Decision Letter 0]

20 Jul 2021

Reviewer #1: In this study, the authors calculated both the direct and indirect costs of hospitalization for severe infection in young infants in Nepal and compared these costs to the maximum limit set by the Government of Nepal’s Free Newborn Care reimbursement. The study addresses an important topic and has been nicely done, and the paper is well-written. I do have some comments and questions for the authors.

Main comments:

1. My primary question concerns the commingling of sepsis with “clinical severe infection.” In the abstract the authors use the term “sepsis,” but in the Methods section state that the terms “sepsis” and “clinical severe infection” are interchangeable. While all sepsis infections are clinical severe infections, all clinical severe infections are not sepsis. Only 21% of the children in the study were culture positive for sepsis. There was, however, quite a difference in the median (IQR) costs for children with and without culture positive sepsis (ll.220-222).

It seems like it would be more accurate, then, to refer to these children as those with “severe infection, including sepsis” with some discussion about the infections of the 80% of children who were not culture-diagnosed with sepsis. In order to justify analyzing these two groups together, the authors should examine whether the two groups are actually similar enough (in characteristics and cost) to be lumped together or should be analyzed separately. Were children who met clinical criteria for sepsis negative by culture? If so, it bears discussing why the authors refer to all infections as “sepsis.”

To inform policy guidelines, the authors should at least do a sensitivity analysis of the two groups separately, since the Government decisions about reimbursements could at some point specify higher reimbursements for culture-diagnosed sepsis and not other non-sepsis infections (even though the costs are also high). I think that a more granular presentation of the data would be the most persuasive.

Thank you for your comment. We apologize for the confusion. We have now removed the word “sepsis” and replaced it with “clinical severe infection”. Study participants were recruited into the clinical trial if they fulfilled inclusion criteria that was an adapted version of the WHO IMNCI strategy to identify sick infants with possible severe bacterial illness. We used findings from a trial to enable recruitment of infants with clinical signs having higher specificity for the diagnosis of possible severe bacterial illness. (African Neonatal Sepsis Trial (AFRINEST) group, Tshefu A, Lokangaka A, Ngaima S, Engmann C, Esamai F, et al. Simplified antibiotic regimens compared with injectable procaine benzylpenicillin plus gentamicin for treatment of neonates and young infants with clinical signs of possible serious bacterial infection when referral is not possible: a randomised, open-label, equivalence trial. Lancet. 2015;385:1767–76.)

While all study participants had recommended screening tests for sepsis, including blood culture, performed at enrolment, treatment was initiated using a standard protocol followed at the study site. The positive blood cultures in 21% of infants did not alter treatment provided to the study infants. 

The clinical trial on which this manuscript is based was carried out to assess efficacy of the intervention in infants with possible severe bacterial illness acquired in the community. (Wadhwa N, Basnet S, Natchu UC, Shrestha LP, Bhatnagar S, Sommerfelt H, et al.; zinc sepsis study group. Zinc as an adjunct treatment for reducing case fatality due to clinical severe infection in young infants: study protocol for a randomized controlled trial. BMC Pharmacol Toxicol. 2017 Jul; 18(1):56.) Our objective of performing blood cultures was to describe the epidemiology of bacterial infections in infants with clinical severe infection and not to inform treatment. We therefore did not think a sensitivity analysis suggested by the reviewer was needed. 

2. The other confusion is with the use of the word “infants” to refer sometimes to all the children in the study (l. 183; also, both age groups are labeled “infants” in Table 1), and sometimes only to the non-neonates (ll. 164-65, l.167). Sometimes the phrase “young infants” is used. In the Results section it is confusing what group constitutes the denominator of the reported percentages. The authors should use these terms consistently throughout the paper to avoid confusion, defining two groups (e.g., specifically defining “neonates” and “infants” as representing different age groups). At times these two groups are analyzed together and at times they are analyzed separately. Again, the authors should be consistent. If the two groups are found to be different enough not to be lumped together, they should be kept separate throughout. This might actually give more precise information than simply adjusting for age in the multivariable model. If the authors feel it is better to combine the groups, they should justify this decision.

Thank you for your excellent suggestions. We have now rectified our terms and use them consistently throughout the manuscript such that neonate refers to age group of 3 - 28 days and young infants to those 29 - 59 days of age. 

We also added data to the baseline table to reflect demographic and clinical characteristics of these two age groups of study participants as suggested (Table 1) and have described the findings (Lines 156 to 163)

We redid the analyses with age as a categorical variable and present our findings in Table 3 and Lines 131 to 132 of the manuscript. 

Other comments:

Introduction:

The majority of the introduction (ll. 52-60) discusses infant deaths in general. This is interesting, but not really necessary. Discussing deaths in infants due to infection would be more relevant to the paper and make the introduction more focused.

Thank you for your suggestions. We have now tried to remove unnecessary sentences and have added findings relevant to the problem

l.68. Perhaps a different word could be used than “schemes” which has a somewhat negative connotation.

We have replaced the term schemes with “plan”.

ll.108-113 Here the authors provide a nice and clear explanation of the calculation of indirect costs

Thank you for your comment

Results

• ll. 165-66 These costs are not normally distributed, so means and sds should not be reported. The correct summaries are in Table 2.

Thank you for your suggestion. We had restricted information on mean cost to the text in order to enable comparison with other studies that have reported means. 

We have kept the information as is, mainly because we received suggestions from another reviewer to report means rather than medians. In fact, we have added a new table of mean costs as Supporting Information (S1 and S2 Tables) as suggested by the reviewer. 

• l.167 Does “with sepsis” refer to all or only those with culture-positive sepsis diagnosis?

The term “with sepsis” included participants with positive and negative culture reports. We have now removed this term to avoid confusion and replaced it “clinical severe infection”, as empirical treatment following a standard protocol was initiated based on clinical signs rather than the culture report which was available only after 48 hours.

• ll.183-85 Please show the results of the analysis for the total costs. The authors’ inclusion of indirect costs (rightfully) in the analysis makes the total costs more relevant than the direct costs; not showing an analysis that includes them implies that they aren’t important (can show both if you want).

We have now added a table showing analysis of total costs in the Supporting information (S3 Table). As results of both analyses were no different, we chose to display the table on analysis of direct costs in the manuscript as we feel this is more relevant for the FNC service which takes only direct medical costs into account. 

• I would argue, based on the R2 values, that the authors should use Model 2 as their final adjusted model.

Thank you for the suggestion. Our results and discussions are based on findings in Model 2

Table 1.

• Refer to age groups by different names (“neonates” and “infants”) here and consistently throughout the paper.

Thank you for the suggestion. We have now categorized into age groups as:

- Infants aged 3 – 28 days = neonates

- Infants aged 29 – 59 days = young infants 

• Education information would be more useful if parental information was combined. Two parents with above primary education are different from one parent with an above primary education and one with no education, or two with no education. I would suggest categorizing them into mutually exclusive groups: (1) neither parent has a formal education (2) one or both parents have a primary education (but no higher) and (2) one or both parents have an above primary education.

Thank you for your suggestion. We recategorized the education information based on feedback provided by you and also performed the regression analysis using this information. However, there was no change in the results and therefore we chose to retain the information on education in Table 1.

Table 3.

• What is the reference category for “Moderately malnourished?” (since this appears to be protective). You might want to reverse the reference group.

Thank you for noticing this. We have now clarified this, and reference group now denotes infants that are not malnourished. 

Discussion

• I am not sure where the USD 122 (l.198) comes from?

This was the exchange rate from 2018 of the mean costs for the study. Since, we have now separated two groups, neonates and young infants, the mean costs have now been revised accordingly in the discussion section. The mean costs are also present in the Supporting information (S1 and S2 table). 

• In comparisons made with other studies, the authors should only compare parallel age groups. In ll.218-220, it sounds like all children (neonates and infants) are being compared to the Sunny study which enrolled only neonates. Again, I think it would be interesting to break out the two age groups in terms of characteristics (Table 1) unless the authors have tested whether there are no differences. The comparisons with other studies made in the discussion would also be less confusing with a more rigorous and consistent delineation between “neonates” and “infants.”

As stated earlier, we have followed the suggestion by the reviewer and made the necessary changes to the results and the discussion section. (Lines 156 to 163, Table 1)

Overall, the authors have done a good job and address an important topic. I do feel that the paper would benefit from more granular analysis (neonates vs. infants; culture-dx sepsis vs. other infections) which might be more useful in informing Government policy. Also, I feel that it is misleading to define all infants as having sepsis, unless the hospital makes diagnoses of sepsis based on clinical diagnoses in the absence of culture-positive diagnosis.

Thank you once again. While we have followed your suggestions to look at the characteristics of participants in the two age groups, we have provided a justification for not analyzing according to reports of blood culture earlier. 

Reviewer #2: Thank you for the opportunity to review the manuscript “Assessment of hospitalization costs and its determinants in young infants with sepsis at a tertiary hospital in Nepal” . My comments are:

In general, all abbreviations should be explained and the perspective of the costs should be mentioned explicitly according to the guidelines of economic evaluations.

Thank you for your comment. We found the following during our search of the literature for guidelines [(2008) Indirect Costs. In: Kirch W. (eds) Encyclopedia of Public Health. Springer, Dordrecht. https://doi.org/10.1007/978-1-4020-5614-7_1685]

1. Definition of Direct Costs: In health economics, the term direct cost refers to all costs due to resource use that are completely attributable to the use of a health care intervention or illness. Direct costs can be split into direct medical costs and direct non-medical costs. Direct medical costs include the cost of a defined intervention and all follow-up costs for other medication and health care interventions in ambulatory, inpatient, and nursing care.

Direct non-medical costs include e. g. transportation costs and additional paid caregiver time. 

In the manuscript we have provided an explanation of how we derived “direct cost” in Lines 107 to 108 and 110 to 111 

Based on the above definition, we have information on direct medical cost and not non-medical costs. 

We have added that this is a limitation of the study in the discussion in Lines 267 to 271

2. Definition of Indirect Costs: In health economics, the term indirect cost refers to all costs to the national economy of the society due to productivity loss. Indirect costs can be due to decreased efficiency, total absence from work through an illness, or premature death. Indirect costs can be estimated using the human capital approach or the friction cost method. Both approaches are based on the assumption that the lost productivity can be valued by the achievable gross income of the employed population.

In the manuscript we have calculated indirect costs using the human capital approach and explain how in Lines 112 to 115 

3. Bed Charge which is another term for hospitalization charges has been included in the direct costs of the manuscript 

Introduction

The introduction reflects the available knowledge about sepsis in the newborn and the importance of the study questions.

Methods

The term “caregivers” is a term of wide comprehension. For me is not clear what the authors mean: parents, relatives or even nurses?

Thank you for your suggestion. We have addressed this comment by now replacing the term caregivers with parent/guardian to avoid confusion. 

It is not clear why the authors did not make a cost benefit or cost effectiveness analysis. I assume that the intervention was not successful. If I am right, this fact should be mentioned in the manuscript.

In this manuscript we describe findings from data collected at one of the sites of a multicenter trial. While enrolment at our site has ended, the recruitment of participants is still ongoing at other sites. Therefore, in this double blinded placebo controlled clinical trial we are not aware of the effects of the study intervention.

We decided to do a retrospective analysis of the costs borne by parents/guardian, for the treatment of infants, meeting criteria for “clinical severe infection”, at a tertiary level public hospital in Nepal. Our aim to calculate average direct medical costs for treatment of clinical severe bacterial infection was to compare this with the maximum limit set by the Government of Nepal in the Free Newborn Care service that has been explained in the introduction. (Lines 85 to 87) 

We did not intend to conduct a cost benefit or cost-effective analyses. 

From my point of view the direct cost calculation is difficult to understand. What does is mean “investigation cost"? Lab cost, labor cost of physicians or what else? Furthermore, I am not sure that the indirect cost is calculated precisely. For example, can it be that newborns or infants were in the hospital for other reasons and sepsis occurred at any point if the hospital stay? This can lead to an overestimation of costs as well as the use of the hospital stay as a proxy for the time lost by the caregivers. Or are in the study only newborns and infants who are admitted to the hospital with diagnosis sepsis from different secondary level hospitals? I think this part of the manuscript needs some clarification for the reader. I am sorry but I am confused.

The “Investigation cost” for the study includes costs for recommended diagnostic tests that were done in study participants with clinical severe infection and as outlined in the study protocol. 

Our study population comprised of infants meeting eligibility criteria for clinical severe infection acquired in the community. The inclusion and exclusion criteria are outlined in the Box 1. As the study site is a tertiary level hospital, some infants may have been referred from secondary level hospitals. However, we excluded those that had been treated with parenteral antibiotics for more than 48 hours and therefore overestimation of the proxy for time lost by the parents/guardians in unlikely. 

Statistical analysis

Statistical analysis is ok.

Results

For table 3 I am missing an explanation in the methods section. Furthermore, what is the rationale for using these models?

One of the study objectives was to identify determinants of increased costs for treatment of clinical severe infection at a tertiary level public hospital in Nepal. (Line 86 to87) 

We have provided an explanation of how we constructed the Models represented in Table 3 in the section on Statistical Analysis in Lines 135 to 148

Discussion

In the discussion section it should be explained why the newest definition of sepsis was not used and why the medicine costs are lower than the investigation costs. That is surprisingly for me. Furthermore, the difference between charges and costs should be discussed. That ist not always the same.

Thank you for your comment. In Nepal, the WHO IMNCI guidelines is recommended for the diagnosis and management of possible severe bacterial illness in young infants. (Physician chart booklet. Integrated Management of Neonatal and Childhood Illnesses. India: World Health Organization, Geneva, UNICEF & Ministry of Health & Family Welfare Govt. of India; 2003). In the study on which this manuscript is based, we used an adapted version of the WHO IMNCI guidelines and recruited infants with a diagnosis of ‘clinical severe infection’. This has been explained in the methods section. (Lines 93 to 95, Box 1) The study was conducted in 2017/2018, whereas the newest Sepsis guidelines endorsed by AAP was published in 2020. (Weiss SL, Peters MJ, Alhazzani W, et al. Surviving Sepsis Campaign international guidelines for the management of septic shock and sepsis-associated organ dysfunction in children. Pediatr Crit Care Med. 2020;21(2):e52–e106.) 

All infants in the study had laboratory and other investigations done following a standard protocol. While medicines were also provided using a standard protocol, the only reason we can think of for this disparity is as follows. The infants (< 2 months old) received treatment with antibiotics for clinical severe infection with calculation of drug doses based on their weight. Therefore, the amount of drug consumed by infants in our study group is likely to be less when compared to adults and older children. 

We have used the term ‘Bed charge’ which also means hospitalization charge and included this in our calculation of direct cost. 

We do not have information on hospital human resources wages and time costs or costs of capital items which is a limitation of our study that we have highlighted in the discussion section. (Lines 276 to 281)

Overall

This is interesting manuscript. Once again, that you for the opportunity to read it.

Reviewer #3: The present topic assesses the treatment costs for neonatal illnesses in tertiary care setting. Undoubtedly the topic is of great interest as healthcare decisions are largely dependent on costs and cost-effectiveness of services/technologies these days. At present there are several issues in the analysis and manuscript. First, the title of the study is ‘Assessment of hospitalization costs and its determinants in young infants with sepsis at a tertiary hospital in Nepal’ which I found a bit misguiding. In classical terms, cost and expenditures are two different concepts. In some instances, expenditures are proxy of costs but more often these are different. In this study authors have done an assessment of patient expenditures on hospital treatment for neonatal sicknesses to serve the purpose of informing FNC package rate which seems incorrect to me. Second, this study does not even comprehensively capture the information on patient expenditures to inform the economic burden of neonatal illnesses though the major heads have been covered. Third, the multivariable analysis is a bit unclear especially for the choice of variables theoretically selected for regression model and missing details how regression was performed.

I think there is a lot scope of improvement in the methodological and analytical part; and therefore, I recommend a major revision. Also, I suggest the authors should present this analysis as assessment of economic burden as a result of neonatal illnesses adding more data (if possible). Specific comments for each section are given below:

Introduction

• Page 3, lines 71-76: The description about the current reimbursement package under the insurance scheme is a bit unclear. Authors mention that the package is expected to cover costs such as investigations, medicines and bed charges for hospitalization and the maximum amount which can be claimed only in case of ICU which includes all A+B+C components. Is the package rate same for normal hospitalization and ICU? Is this package rate similar or differential across the public and private hospitals? Is there a scrutiny on claims raised by the hospitals? Any reflections from claims data about total claims settled under FNC package (or burn out ratio if possible)? More details would be useful to understand the context and to evaluate whether the current costing study serves the purpose of informing the required revisions in reimbursement rates of FNC 

Thank you for your comments. We have now revised the whole paragraph on the Free Newborn Care service offered by the Govt of Nepal in the introduction. (Lines 65 to 80)

We hope the information provided is clear now.

• Page 4, lines 78-79: “Moreover, this package does not take into account the indirect

out of pocket costs that caregivers of the sick neonates have to bear.” I doubt if we have any such precedence of package rates considering the indirect costs. Generally, the package rates tend to cover the treatment costs that too preferably the cost of provisioning of a particular service. In some instances, direct non-medical costs such as expenses for travel etc are covered but productivity losses are not covered as these neither fall within the scope of packages nor systems have enough resources to account for it.

Thank you for your comment. We agree and have removed the line from the introduction. 

Methods

• My main concern is authors study the patient expenses incurred for treatment of sick neonates in tertiary setting and claim that the aim of the present study is to inform the planning of budget and basis for revision in FNC package rate for neonatal care. I think the methodology employed is not appropriate to address the policy question in view of following arguments. Firstly, the patient expenditures for neonatal illnesses at one side are important to be assessed to understand the economic impact in terms of financial hardship to households but certainly not an appropriate basis for setting the prices of healthcare packages. Second, patient/household expenditures are subject to high variation and cost of medicines and diagnostics drives this variation. Even if the medicines, diagnostic services etc. were available from the hospital itself, the charges at public hospital are supposed to be subsidized and therefore, what patient pay does not adequately represent the cost of care in true economic terms. Moreover, the patients are not charged for time of human resource (Doctors, paramedical staff etc.) involved in treatment and care which accounts for around 40-70% of total cost of service and the capital resources consumed for service delivery. Third, the most appropriate way to inform such policy decisions is to estimate the value of resources consumed for delivery of a service that truly represent the cost of care. Now, this could be done employing the standard methodologies for healthcare costing (Bottom up, normative, mix methodology etc.). The normative costing approach seems to be perfect here as there are standard treatment protocols for neonatal care under IMNCI. There is some practical guidance available on the subject. (Özaltın A, Cashin C. Costing of health services for provider payment. A practical manual based on country costing challenges, trade-offs, and solutions. Arlington: Joint Learning Network for Universal Health Coverage. 2014; Translating Research to Policy: Setting Provider Payment Rates for Strategic Purchasing under India’s National Publicly Financed

Health Insurance Scheme)

Thank you for your very good suggestions. 

We have documented payments made to the hospital for the diagnostic investigations and bed charges and for medicines bought for the treatment by the parents/guardians of study participants. This was possible because these payments were reimbursed after they provided us with the bills that we kept. We tried but were unable to derive reliable estimates for capital costs of items and other items suggested by you. We are therefore unable to carry out the exercise suggested by you and have discussed this as limitations of our study methodology in the discussion section. (Lines 276 to 281)

• No information on perspective used for costing though it seems to be patients’ perspective which mismatches with policy question to be addressed.

We realize that our aim to inform policy with the limited information we have is not justifiable. We have now changed it in the introduction section and have pointed out that our findings may help the successful implementation of the FNC service by providing evidence on the direct costs for treatment. (Lines 85 to 86)

• Statistically, median as a measure of average may be justified as it is least affected with high variations compared to mean but for practical purposes mean is a better choice specifically for reimbursement decisions. We must let the average get influenced by the variation (if this is genuine variation). The literature also suggest that the use of mean is quite popular in the costing studies specifically for informing budgetary and planning decisions.

Thank you for your suggestion. We have projected the means of the various cost components in S1 and S2 Tables in the Supporting information section and have mentioned a few lines on means costs in the text in the Results section (Lines 169 to 171)

As one of the other reviewers has suggested we display median and not means we have therefore kept the Table 2 with median costs in the Results section.

• Human resource wages, cost of capital items and cost of administration/support services not accounted in estimation of costs no matter whether patients are paying for it or not if the overarching goal is to generate basis for FNC package rate. Its exclusion is only justified if authors claim the purpose of study to be assessment of patient out-of-pocket expenditures associated with neonatal treatment and care. But currently the latter argument does not hold completely true as the travel and other costs are absent.

Yes we agree that our calculated costs do not include components mentioned by the reviewer and this is a limitation of our study, which we have explicitly discussed. (Lines 276 to 281 in discussion section). Our aim to provide a basis for the FNC package stems from the fact that it is not clear why a maximum limit of NPR 8000 was set. Also our findings clearly show that this amount for care in a tertiary care public hospital is inadequate and confirms the findings of the study on FNC service in Nepal.(Shrestha G, Paudel P, Shrestha PR, Jnawali SP, Jha D, Ojha TR, Lamichhane B. Free Newborn Care Services: A New Initiative in Nepal. J Nepal Health Res Counc. 2018 Oct 30;16(3):340-344. PMID: 30455497)

• Authors have converted the costs from Nepalese rupee to USD using the exchange rate of 2018. As we are already in 2021, so I suggest authors to consider adjusting the estimates to current period for better utility.

We agree that these are older rates. We have now kept the mean direct cost after accounting inflation till 2020 in Lines 206 to 207 

• Page 6, lines 131-135: Authors have considered including following independent factors as determinants in multivariable regression Age of infant, gender, nutritional status, place of delivery, mother’s age, father’s age, mother’s education, father’s education, symptoms of illness (fever, lethargy, stopped feeding well), signs of clinical severe infection (severe chest indrawing, febrile or hypothermia, movement only when stimulated), signs of critical illness (grunting, nasal flaring, convulsions and no movement at all) and treatment failure.

In my opinion, the independent factors (as determinants/predictors) are included in the regression based on some theoretical relevance. Now, finding the association between these factors with outcome of interest does not guarantee the presence of any causal relationship which is a prerequisite in case of determinants. I doubt if there is any theoretical relevance of including factors like age of father and mother; place of delivery (what this stands for? Seems to be place of childbirth? If yes, what are categories within it? Home deliveries within this might be of greater importance). Also, there are too many syndromic variables. Why not to combine these and could be used as dummy variables. Otherwise, there will be a singularity issue in the regression even if the authors have taken care of multicollinearity issues.

We totally agree that this will not imply any causality as this is an observational study. We are therefore only looking for associations. As per the reviewer’s suggestion we have not included Father’s education in the analyses and have also renamed category ‘place of delivery’ to place of birth as home/hospital 

We have also changed the variable age from continuous to dichotomous, now all variables are dichotomous (categorical/dummy) 

• Page 6, lines 139-140: Please explain manual stepwise approach. I guess author meant guided stepwise regression. If yes, do we include or exclude the variables in regression at every step? Is it done for a single variable every time?

Thank you for your suggestion. We have now added the step wise approach to the statistical analysis section (Lines 142 to 148)

Results

• Page 8, Table 2: How come the average of independent components of costs i.e., bed charges, investigations and medicines not equal to average of total direct costs?

We have put medians in Table 2 not means. 

• I am not sure how the independent variables were introduced into the regression model. Some of variables seems to be on continuous scale whereas some of the variables are of nominal or ordinal nature such as place of delivery, education of parents, signs and symptoms etc. Use of dummy variables is generally recommended for multivariable regression in case of categorical variables. Please clarify.

We have changed age to a categorical variable to denote neonates versus young infants. Now all our independent variables are dichotomous (catergorical/dummy). 

• I think authors should have given the breakdown within each section i.e., medicines and investigations as it is important to understand what are the major drivers of costs within each subhead. For example, if a particular drug or a diagnostic test is predominantly determining the cost in respective subheads then there is a clear direction for policy makers for targeted action needed. This particular evidence will highlight the urge for strategic purchasing focusing on value-based pricing.

Thank you for your suggestion. In this clinical trial, all study infants were managed according to clinical protocol and therefore significant variation between investigations and medicines is unlikely. Moreover the FNC package does not cover individual costs for medicines and investigations but for the service provision according to type and in different levels of health institutions as a whole. 

Discussion

The discussion part starts very well but losses the focus in between with too much comparison with other studies. I suggest the authors should curtail this part with most relevant comparisons. Also, the discussion should be revised in view of methodological comments given in the previous sections and should add a focused para on policy implications which should cover all the aspects suggested in previous comments.

Thank you for your excellent suggestion. We have now rewritten various subsections of the discussion and have tried to add more relevant arguments. 

Minor comments

• In the references 3, 6, 9 etc. the access date would be more relevant instead of date of citation.

- The access date is the cited date in the reference. 

• References 11, 12, 19 and 22 are Government reports, so the weblink and access date should also be provided.

- The weblink has been added along with the cited date which is same as the access date

• The WHO reference after reference 19 had not been numbered. Please check the in text citations also for consistency.

The reference had been repeated, we have now corrected this.

---

## [Editor Report · Decision Letter 1]

8 Sep 2021

PONE-D-20-32967R1Assessment of hospitalization costs and its determinants in infants with clinical severe infection at a tertiary hospital in NepalPLOS ONE

Dear Dr. Basnet,

Thank you for submitting your revised manuscript to PLOS ONE. After careful consideration, we feel that the revised manuscript has improved significantly but still there are points which needs to be addressed. Therefore, we invite you to submit a revised version of the manuscript that addresses the points raised during the review process. The additional comments are given below.

We look forward to receiving your revised manuscript.

Kind regards,

Pankaj Bahuguna, Ph.D.

Academic Editor

PLOS ONE

Journal Requirements:

Additional Editor Comments (if provided):

Additional Comments on Revised Manuscript

1. Based on previous comments to enhance clarity on sepsis and clinical severe infection, the authors may need to add more details for relation between sepsis and clinical severe infection as the abstract starts with sepsis (line 26) and in the aims of the study mentions estimation of costs of clinical severe infection (line 30). Same holds for first para in introduction.

2. The title should include public sector tertiary hospital if this was the case as mentioned in line 44.

3. Lines 121 to 123 mention about various cost centres that are directly involved in provision of care to infants/neonates. It would be useful to have granular details of unit cost by these cost centres in the supplement file since 74% of the recruited patients were in paying wards and private cabins.

4. Line 160-161: It would be useful to know the average length of stay in different cost centres-wards, paying wards, private cabins, ICU settings for reimbursement rates purpose to arrive at per bed day cost in these settings. This could be added in the supplement file.

5. Line 266 could be rephrased to direct medical costs as authors did not collect information of direct non-medical costs.

6. Supplementary table S3 with geometric means can be excluded. This is too much information for readers which will confuse them. Moreover, authors have already provided median estimates in the main manuscript and I assume median and GM to be close.

7. Table 2, page 8: I suggest the 4 and 5 digits median cost estimates should not be shown in decimals. It is better if the cost estimates upto 3 digits are shown in decimals (that also only upto one decimal place). Same suggested for figure 2. Also, give the full form of IQR under the table 2.

8. The font size of text varies at several places such as reference 20 and; page 3 and 4. Also, there seems to be inconsistency in text spacing. The manuscript should be read carefully to check the same.
---

## [Author Response · Author response to Decision Letter 1]

21 Oct 2021

1. Based on previous comments to enhance clarity on sepsis and clinical severe infection, the authors may need to add more details for relation between sepsis and clinical severe infection as the abstract starts with sepsis (line 26) and in the aims of the study mentions estimation of costs of clinical severe infection (line 30). Same holds for first para in introduction.

- Thank you for your suggestion. We have added a few lines in the abstract (Lines 30 to 31) and also to the introduction section (Lines 60 to 70) explaining this relationship and hope that it is clear now . This required the addition of 4 extra citations that have been added to the reference section.

2. The title should include public sector tertiary hospital if this was the case as mentioned in line 44.

- We have now added the suggested, ‘public’ in the title

3. Lines 121 to 123 mention about various cost centres that are directly involved in provision of care to infants/neonates. It would be useful to have granular details of unit cost by these cost centres in the supplement file since 74% of the recruited patients were in paying wards and private cabins.

We are unable to provide granular details of unit cost, disaggregated by the ‘cost centres’. However, we have prepared a supplementary table (S3 Table) with details of cost per day of each cost centre. 

 We would also like to clarify that 74% admitted to other wards (private and paying) represents 50 out of 68 young infants and not all enrolled study participants. The remaining (138 out of 206) comprised of neonates admitted to Neonatal Intermediate wards and a Neonatal Intensive Care unit.

 4. Line 160-161: It would be useful to know the average length of stay in different cost centres-wards, paying wards, private cabins, ICU settings for reimbursement rates purpose to arrive at per bed day cost in these settings. This could be added in the supplement file.

-Thank you for your suggestion. We agree that it would be useful to know the average length of stay in different cost centres for reimbursement purpose to arrive at per bed day cost. As stated in the earlier response we are unable to provide this data.

5. Line 266 could be rephrased to direct medical costs as authors did not collect information of direct non-medical costs.

- Thank you for pointing this out. We have rephrased as suggested.

6. Supplementary table S3 with geometric means can be excluded. This is too much information for readers which will confuse them. Moreover, authors have already provided median estimates in the main manuscript and I assume median and GM to be close.

- We have now removed geometric mean from the Supplementary table

7. Table 2, page 8: I suggest the 4 and 5 digits median cost estimates should not be shown in decimals. It is better if the cost estimates upto 3 digits are shown in decimals (that also only upto one decimal place). Same suggested for figure 2. Also, give the full form of IQR under the table 2.

- We have now revised the number of decimals as suggested

8. The font size of text varies at several places such as reference 20 and; page 3 and 4. Also, there seems to be inconsistency in text spacing. The manuscript should be read carefully to check the same.

- We have now corrected the font size in the manuscript and modified the spacing according to requirements of the journal.

---

## [Editor Report · Decision Letter 2]

4 Nov 2021

Assessment of hospitalization costs and its determinants in infants with clinical severe infection at a public tertiary hospital in Nepal

PONE-D-20-32967R2

Dear Dr. Basnet,

We’re pleased to inform you that your manuscript has been judged scientifically suitable for publication and will be formally accepted for publication once it meets all outstanding technical requirements.

Kind regards,

Pankaj Bahuguna, Ph.D.

Guest Editor

PLOS ONE
---

## [Editor Report · Acceptance letter]

17 Nov 2021

PONE-D-20-32967R2 

Assessment of hospitalization costs and its determinants in infants with clinical severe infection at a public tertiary hospital in Nepal 

Dear Dr. Basnet:

I'm pleased to inform you that your manuscript has been deemed suitable for publication in PLOS ONE. Congratulations! Your manuscript is now with our production department. 

Kind regards, 

on behalf of

Dr Pankaj Bahuguna 

Guest Editor

PLOS ONE